# Optimization of 2024-T3 Aluminum Alloy Friction Stir Welding Using Random Forest, XGBoost, and MLP Machine Learning Techniques

**DOI:** 10.3390/ma17071452

**Published:** 2024-03-22

**Authors:** Piotr Myśliwiec, Andrzej Kubit, Paulina Szawara

**Affiliations:** 1Department of Materials Forming and Processing, Rzeszow University of Technology, al. Powst. Warszawy 8, 35-959 Rzeszów, Poland; 2Department of Manufacturing and Production Engineering, Rzeszow University of Technology, al. Powst. Warszawy 8, 35-959 Rzeszów, Poland; akubit@prz.edu.pl; 3Doctoral School of Engineering and Technical Sciences, Rzeszow University of Technology, al. Powst. Warszawy 12, 35-959 Rzeszów, Poland; p.szawara@prz.edu.pl

**Keywords:** friction stir welding (FSW), aluminum alloy 2024-T3, FSW tool, machine learning, random forest, XGBoost, multilayer perceptron (MLP-ANN), hyperparameter optimization, grid search, response surface methodology (RSM), welding parameter optimization

## Abstract

This study optimized friction stir welding (FSW) parameters for 1.6 mm thick 2024T3 aluminum alloy sheets. A 3 × 3 factorial design was employed to explore tool rotation speeds (1100 to 1300 rpm) and welding speeds (140 to 180 mm/min). Static tensile tests revealed the joints’ maximum strength at 87% relative to the base material. Hyperparameter optimization was conducted for machine learning (ML) models, including random forest and XGBoost, and multilayer perceptron artificial neural network (MLP-ANN) models, using grid search. Welding parameter optimization and extrapolation were then carried out, with final strength predictions analyzed using response surface methodology (RSM). The ML models achieved over 98% accuracy in parameter regression, demonstrating significant effectiveness in FSW process enhancement. Experimentally validated, optimized parameters resulted in an FSW joint efficiency of 93% relative to the base material. This outcome highlights the critical role of advanced analytical techniques in improving welding quality and efficiency.

## 1. Introduction

Friction stir welding (FSW) was invented and patented by Wayne Thomas at The Welding Institute (TWI) in the United Kingdom in 1991 [1]. The method utilizes a non-consumable rotating tool with a specially designed shoulder and pin to generate heat through frictional forces [2]. As the tool traverses along the joint line, the softened material is mechanically stirred, resulting in the formation of a solid-state bond [3].

In the current context of advanced manufacturing technologies, friction stir welding (FSW) serves as an example of engineering innovation transformed by the introduction of machine learning (ML) and artificial neural networks (ANN). These tools are reshaping the approach to optimizing, controlling, and forecasting FSW processes, aligning with the broader industrial goals of increasing efficiency and ensuring weld quality.

The use of machine learning (ML) techniques in the optimization of the friction stir welding (FSW) process has attracted considerable attention in recent years. Researchers have applied various ML methods, such as artificial neural networks (ANNs), support vector machines (SVMs), and predictive modeling, to enhance the FSW process. For example, Nasir and Solyali [4] discussed the applications of ANN and SVM in optimizing FSW, emphasizing their high prediction accuracy and efficiency in process optimization. Similarly, Sarsılmaz and Kavuran [5] found that ANN models surpassed SVM techniques in optimizing FSW parameters to improve the tensile strength and hardness of welded joints.

The significance of ML and ANN for process optimization in FSW has been extensively examined in the recent literature, with a study by Prabhakar et al. [6] emphasizing the influence of key FSW parameters such as rotational speed, tilt angle, traverse speed, and tool profile geometry on joint quality. Their research explored the interaction between data acquisition through integrated sensors and analysis using AI algorithms, resulting in significant enhancements in weld quality. This study highlighted the combination of data-driven optimization and empirical evaluation in contemporary FSW processes, underscoring the crucial role of ML in unraveling the complexities of welding parameters to achieve high-quality joints.

Attention has been directed not only towards the refinement of welding parameters but also towards the predictive analytical capabilities of ML. The employment of diverse regression models and cross-validation techniques by Anandan and Manikandan [7] has introduced a structured approach to forecasting FSW results. Moreover, this capacity for prediction is expanded through hybrid modeling employing various ML algorithms, as investigated by Ye et al. [8]. This research offers insights into the mechanical properties and behaviors of materials undergoing FSW, demonstrating the extensive applicability of ML beyond standard process parameters and into the sphere of advanced materials science.

The microstructural evolution of materials during FSW further benefits from the simulation and predictive capabilities of ML. Silva et al. [9] conducted a detailed investigation to model the FSW process on stainless steel joints and elucidate the structural changes. This approach, pointing towards the integration of simulation tools with ML predictions, is set to be instrumental in defining the process conditions that yield desired joint qualities, harnessing computational tools to anticipate and plan for manufacturing outcomes.

The frontier of ensemble machine learning methods, as demonstrated by Matitopanum et al. [10], offers a leap forward in predicting the ultimate tensile strength of FSW joints. These models present an innovation in forecasting, merging complex algorithms such as Gaussian process regression and support vector machines, and outclassing traditional statistical approaches in accuracy and predictive performance.

Artificial neural networks (ANNs) have proven to be equally revolutionary. The research by Essa et al. [11] employed sophisticated ANN models to forecast the impact of tool eccentricity on the mechanical properties of welded aluminum alloy, exemplifying the precision with which ANNs can target specific FSW parameters to predict and optimize welding outcomes. These results showcase the utility of ANN in achieving a granular level of control over the welding process, tuning it to the material type and specific welding conditions.

Elsheikh [12] explores the extensive applications of ML in FSW, encompassing the prediction of joint characteristics, real-time process monitoring, and diagnostics. This research advocates for the thorough integration of ML into the FSW process, transforming it from a mere material joining technique to a sophisticated procedure that greatly benefits from the analysis of real-time data and applied artificial intelligence. Furthermore, Lacki et al. [13] utilized ANNs to formulate process parameters for aluminum joint FSW. Their findings highlight the effectiveness of ANN in FSW, establishing computational artificial intelligence as a crucial element for improving not only the process’s efficiency but its consistency as well. Predictive models provide valuable guidance in choosing the best FSW process parameters for different joint types.

Kubit et al. [14] utilized Design of Experiments (DoE) to optimize parameters in friction stir welding for aluminum alloys, focusing on both the strength and consistency of welds. This study highlights the shift towards a balance of multiple objectives in welding outcomes, showcasing the role of machine learning in enhancing process efficiency across the FSW manufacturing sector.

Certainly, the integration of machine learning (ML) and artificial neural network (ANN) techniques for the optimization of the friction stir welding (FSW) process is reflective of a broader trend where such technologies are harnessed to enhance efficiency across various industries. The study by Li et al. [15] underlines the pervasive influence of data mining, a key component of ML, in sectors like chemistry and materials engineering. The authors propose a generalized data-mining strategy that can be applied to optimize processes in a variety of fields, highlighting the versatility of ML approaches.

Devikanniga et al. [16] discuss the significant role played by meta-heuristic optimization algorithms inspired by nature in enhancing the efficiency of ANNs. Such hybrid networks have shown marked improvements in classification and prediction tasks, edging out standard ANNs in terms of performance. Their review points out that such advancements in ANN optimization can directly benefit FSW process control by allowing for better adaptiveness to changing parameters and optimization for specific welding scenarios.

Other recent works contribute to the understanding of ML’s capabilities in FSW. For instance, Nadeau et al. [17] evaluated various machine learning approaches, such as principal component analysis, K-nearest neighbor, multilayer perceptron, SVM, and random forest methods, applied to friction stir welding. Their study correlates welding process variables (e.g., rotational speed, travel speed, and axial force) with a defect index, showing the potential of ML in quality control for FSW.

In line with improving predictive models, Verma et al. [18] also tackled the prediction of tensile behavior in FS-welded AA7039 using ML, reinforcing the critical role of ML in achieving predictive accuracy that can anticipate the tensile outcomes of welded materials. The study by Guan et al. [19] suggests a direction toward force data-driven ML for identifying defects in FSW, an application that may revolutionize quality assurance by automating the detection of FSW flaws.

Completing the perspective of analytics in FSW, Dorbane et al. [20] explore deep learning methods to forecast the mechanical behavior of FSW aluminum sheets, thus expanding the methodological toolkit for predicting and ensuring the quality of FSW outputs. Supporting this view, Tien Dat et al. [21] utilized ML to predict temperature distribution in FSW, a critical factor for weld quality. They developed a numerical model with the potential to accurately simulate temperature variations, enhancing the sustainability of the FSW process.

In a study focusing on the challenges in detecting FSW defects, Wahab et al. [22] explore the substantial influence of FSW process parameters on joint quality in aerospace-grade aluminum alloys. They introduce a novel empirical force index to identify optimal welding parameters. This type of index could be enhanced by integrating ML algorithms to analyze and predict which combinations of process parameters are likely to yield defect-free welds.

Deep learning and ANN have extended their reach into various sectors, including FSW, where the opportunity to leverage these technologies can lead to greater efficiency and effectiveness in processes. A review by Wang and Luo [23] talks about the optimal design of neural networks based on field-programmable gate array (FPGA) technology. While not directly related to FSW, the implications of such studies indicate that FPGAs could be used to accelerate ANN computations in FSW scenarios, ensuring faster and more efficient process optimization.

Sakthivel et al. [24] address monitoring FSW using a vision system paired with an ML algorithm. Their approach to visual monitoring, combined with ML’s predictive ability, could revolutionize the way FSW processes are controlled and managed, leading to real-time quality assurance and potentially alleviating the issues of weld defect formation by adjusting welding parameters on the fly to maintain optimal conditions.

These studies collectively indicate that machine learning and artificial neural networks are critical in advancing FSW technology, not just in predicting outcomes but also in optimizing the process parameters to achieve better joint quality. The advancement of such techniques showcases a transformative period where traditional models are being replaced by intelligent systems capable of understanding complex patterns and making more accurate predictions relevant to FSW and beyond.

In this study, two machine learning methods (RF and XGB) and the MLP-ANN neural network method were applied to forecast the ultimate tensile strength (UTS) of FSW joints. Additionally, the models’ hyperparameters were tuned using grid search optimization. The technological parameters applied, and the experimental results obtained from the friction stir welding (FSW) process were used to create predictive models of ultimate tensile strength (UTS) using machine learning (random forest and XGBoost) and artificial neural networks (MLP-ANN). The required scripts were developed in Python, utilizing the relevant libraries. The constructed ML and ANN models were employed to predict UTS based on input parameters (tool rotation and welding speed) and to search for optimal parameters to achieve the highest possible UTS values, approaching those of the base material. The search encompassed the range of studied input parameters and extended beyond this range through extrapolation.

The article presents an innovative approach to optimizing FSW parameters for aluminum alloys using ML and ANN, achieving high prediction accuracy and significantly improving weld quality. It stands out for employing advanced analytical techniques to enhance joint strength, offering substantial benefits to industries requiring precise and reliable welding. A distinguishing feature of this work is the successful training of neural networks on a very limited amount of input data, demonstrating the efficiency and effectiveness of the developed models in scenarios with constrained data availability. This aspect underscores the adaptability and potential of ML and ANN applications in optimizing welding processes, even when extensive datasets are not accessible.

## 2. Materials and Methods

The aim of this study was to utilize the friction stir welding (FSW) technique for butt welding AA2024-T3 aluminum sheets with a thickness of 1.6 mm along the rolling line, spanning a length of 180 mm, employing a Makino PS95 machine. Table 1 presents the parameters of a commercially available tool utilized in the process. The preliminary welding parameters set a tool rotation speed of 1200 rpm and a welding speed of 160 mm/min, which constituted the central point of the experimental design. This was based on preliminary research into the FSW process. The welds produced under these conditions were devoid of defects and exhibited a strength of approximately 85% relative to the base material. To further investigate the possibility of enhancing the strength of the FSW joints, an examination around the central point was initiated by adding an additional 8 factorial points. Consequently, a 3 × 3 factorial experiment plan was developed, leading to a total of 9 welds executed under specific technological parameters as detailed in Table 2. The assessment of mechanical properties was carried out using a Zwick/Roell Z 100 universal testing machine (ZwickRoell, Ulm, Germany), following the ISO 6892:2020 standard [25]. For every set of technological parameters, 5 samples were prepared. The outcomes are summarized in Table 2, and the distribution of these results is depicted in a box plot with whiskers in Figure 1, providing a visual representation of the data’s variability and central tendency.

### Statistical Analysis of Experimental Data

The exploratory analysis of the experimental dataset began with a histogram inspection (Figure 2), which did not unveil the characteristic bell-shaped curve indicative of a normal distribution, suggesting potential deviations. Subsequent computations of skewness and kurtosis produced values signaling leftward asymmetry (−1.147) and a moderately leptokurtic distribution (0.753), respectively.

To address the data’s non-normality, a Box–Cox transformation was applied, resulting in transformed values with an optimal lambda parameter of approximately 5.74. Post-transformation analysis through a new histogram (Figure 3) highlighted an altered distribution shape, which, despite skewness correction, failed to exhibit the essential bell curve of a normal distribution, suggesting that the data might not be perfectly normalized.

Further scrutiny employing a Q–Q plot (Figure 4) revealed many data points aligning closely with the reference line for a normal distribution, primarily in the central plot region. However, deviations at the extremes indicated the presence of heavy tails that the Box–Cox transformation could not fully normalize.

Subsequently, a comprehensive ANOVA was conducted using tool rotation speed and welding speed as independent variables, revealing their significant impact on ultimate tensile strength (UTS), as shown in Table 3. This analysis underscored the substantial influences of welding speed (*p* < 0.0001) and its interaction with tool rotation speed (*p* < 0.0001), demonstrating a pronounced synergistic effect on UTS. While tool rotation speed alone did not emerge as a significant factor (*p* = 0.0885), suggesting a nuanced influence on UTS, its quadratic term was highly significant (*p* < 0.0001), indicating a complex, nonlinear relationship with UTS. Conversely, the quadratic term for welding speed lacked statistical significance (*p* = 0.3033), implying a predominantly linear effect on UTS.

The significant F-value for lack of fit (28.09, *p* < 0.0001) indicated that the model did not capture all data variations perfectly, pointing towards potential unaccounted factors or interactions. Despite this, the low pure error reflected high experimental reproducibility. The ANOVA highlighted the critical interplay between tool rotation speed, welding speed, and their quadratic interactions with UTS, suggesting avenues for model refinement to encompass the full spectrum of UTS influences.

Given the observations and the incomplete achievement of a satisfactory normal distribution fit, the study has turned its attention towards machine learning methodologies for further analysis. Methods such as random forest, XGBoost, and multilayer perceptron artificial neural networks (MLP-ANN) were identified as potent tools for modeling the data’s complex, nonlinear relationships. Known for their robustness to non-normal distributions, these approaches offer valuable insights beyond traditional statistical techniques’ capabilities, marking them as a focal point of this study to enhance the understanding and prediction of UTS behavior.

Figure 5 outlines the methodology utilized for optimizing friction stir welding (FSW) parameters using machine learning techniques. It demonstrates the workflow for three different predictive models: random forest (RF), XGBoost, and multilayer perceptron (MLP). For each model, the process begins with input parameters, including tool rotation speed (S in rpm), welding speed (F in mm/min), and ultimate tensile strength (UTS in MPa). The dataset is divided into a training set and a testing set.

Initial parameter settings for each model are established, followed by the tuning of hyperparameters using grid search to find the optimal settings. This tuning is aimed at minimizing prediction error as measured by various metrics, such as mean squared error. Once the optimal model parameters are identified, the models are trained and then validated through testing.

The testing and validation phase evaluates the models’ performance using metrics such as R^2^, MAE, MAPE, MSE, RMSE, and accuracy to ensure that the models can reliably predict the FSW parameters that result in the best UTS for welded joints. The final output of this process is the set of optimized welding parameters (S and F) that yield the highest possible UTS, which is crucial for ensuring the strength and integrity of welded joints in industrial applications.

## 3. Methodology of the ML and ANN Models

Two ML models and one MLP-ANN were prepared. From the machine learning methods, the random forest method, hereafter referred to as RF, was selected. The next method is XGBoost (eXtreme Gradient Boosting), hereafter referred to as XGB. The input parameters (independent) are tool rotation and welding speed. The output parameter (dependent) is the UTS strength of the joint. Table 2 shows 45 sets of parameters, which were divided into training and testing sets.

### 3.1. Model Evaluation Metrics

The implementation of all models was programmed in Python using the following machine learning libraries: Scikit-learn and XGBoost. To validate the performance of the optimized models, six key indicators were introduced, including the coefficient of determination ($R^{2}$), mean absolute error (MAE), mean absolute percentage error (MAPE), mean squared error (MSE), root mean squared error (RMSE), and accuracy [26].
(1)$$R^{2}=1-\frac{\sum_{i=1}^{n}{\left(y_{i}-\hat{y_{i}}\right)}^{2}}{\sum_{i=1}^{n}{\left(y_{i}-\bar{y_{i}}\right)}^{2}}$$
(2)$$MAE=\frac{1}{n}\;\sum_{i=1}^{n}\left|y_{i}-\hat{y_{i}}\right|$$
(3)$$MAPE=\frac{100\%}{n}\sum_{i=1}^{n}\left|\frac{y_{i}-\hat{y_{i}}}{y_{i}}\right|$$
(4)$$MSE=\frac{1}{n}\sum_{i=1}^{n}{\left(y_{i}-\hat{y_{i}}\right)}^{2}$$
(5)$$RMSE=\sqrt{\frac{1}{n}\sum_{i=1}^{n}{\left(y_{i}-\hat{y_{i}}\right)}^{2}}$$
(6)ACCURACY=100%−MAPE
where n is the number of observations (or data points)$,\;{\;y}_{i}$ is the actual value, $\hat{y_{i}}$ is the predicted value by the model, and $\bar{y_{i}}$ is the average of all actual values y. Furthermore, for each model, comparison plots of the model’s predictions against the actual values and learning curve plots were created.

### 3.2. Random Forest Algorithms

The random forest (RF) model operates as an ensemble learning technique that combines the predictions from multiple decision trees to make more accurate and stable predictions than any single decision tree could on its own. RF builds a collection of decision trees during the training process. Each tree is constructed using a random subset of the training data (bootstrap sample) and at each split in the tree, a random subset of the features is considered. This randomness helps make the model more robust and less prone to overfitting.

For regression tasks, each tree predicts a continuous value, and the final output is the average of all the tree outputs. The prediction of the random forest model, $\hat{y}$ for a regression problem is the average of the predictions from all the individual trees:(7)$$\hat{y}=\frac{1}{n}\sum_{i=1}^{n}T_{i}\left(x\right)$$
where $T_{i}\left(x\right)$ is the prediction of the i-th tree. random forest also provides insights into the importance of each feature in making predictions. The importance of a feature can be measured by calculating the decrease in model accuracy or impurity (Gini impurity for classification, variance for regression) when the feature’s values are permuted across the trees [27].

Advantages include high accuracy due to averaging predictions from multiple trees, robustness to overfitting, especially with a large number of trees, and the ability to handle missing values and maintain accuracy even when a large proportion of the data are missing. Limitations involve increased complexity and computational cost with the number of trees and being less interpretable compared to a single decision tree [28,29,30].

### 3.3. XGBoost Algorithm

XGBoost (eXtreme Gradient Boosting) is an advanced implementation of gradient boosting algorithms, designed for speed and performance. It is a highly efficient and scalable end-to-end tree boosting system. XGBoost’s objective function consists of two parts: the loss function and the regularization term. The loss function measures how well the model’s predictions match the actual data. For a set of n predictions and corresponding actual values, the loss function can be represented as $L\left(\theta \right)$ where θ represents the parameters of the model. The regularization term helps to control the model’s complexity, preventing overfitting.
(8)$$Obj\left(\theta \right)=L\left(\theta \right)+\Omega \left(\theta \right)$$
where $L\left(\theta \right)$ is the loss function and $\Omega \left(\theta \right)$ is the regularization term.

XGBoost improves upon the concept of gradient boosting by sequentially adding predictors (trees), where each new tree corrects errors made by the previously trained trees. The model uses a gradient descent algorithm to minimize the objective function. For a given iteration t, the model updates the predictions based on the negative gradient of the loss function evaluated with the current predictions [31].
(9)$${\hat{y}}^{\left(t\right)}={\hat{y}}^{\left(t-1\right)}+\eta \cdot \sum_{k=1}^{K}f_{k}\left(x\right)$$
where ${\hat{y}}^{\left(t\right)}$ is the prediction at iteration t, η is the learning rate, and $f_{k}\left(x\right)$ represents the k-th tree’s contribution.

XGBoost includes both L1 (Lasso regression) and L2 (Ridge regression) regularization terms in its objective function, which are denoted as $\Omega \left(f\right)$ for a tree f. This regularization term is a key feature that differentiates XGBoost from other gradient boosting methods, making it less prone to overfitting.
(10)$$\Omega \left(f\right)=\gamma T+\frac{1}{2}\lambda \sum_{j=1}^{T}\omega_{j}^{2}$$
where T is the number of leaves in the tree, $\omega_{j}$ is the score on the j-th leaf, γ is the complexity control on the number of leaves, and λ is the L2 regularization term on the leaf weights.

XGBoost automatically learns the best direction to handle missing values. During training, if a split point includes missing values, the model learns whether to categorize these values to the left or right branch, enhancing its ability to handle missing data effectively. Unlike traditional gradient boosting methods that stop splitting a node when no further gains can be made, XGBoost grows the tree to its maximum depth and then prunes back the branches that contribute little to the overall prediction, using the gain from the objective function as the criterion [32,33].

### 3.4. Multilayer Perceptron (MPL-ANN) Algorithm

The multilayer perceptron (MLP) is a type of artificial neural network that is foundational to deep learning. It consists of at least three layers of nodes: an input layer, one or more hidden layers, and an output layer. MLP utilizes a supervised learning technique called backpropagation for training. The multilayer perceptron (MLP) model consists of three main types of layers that work together to process the input signal and produce the final output. It begins with the input layer, which receives the input signal to be processed. This signal is then passed through one or more intermediate layers, known as hidden layers. In these hidden layers, computations are performed by neurons that process the inputs and pass their output to the next layer. The complexity and capabilities of the MLP model are determined by the number of these hidden layers and the number of neurons within each of these layers. Finally, after passing through all the hidden layers, the signal reaches the output layer. It is in the output layer that the final output of the network is produced, representing the result of processing the input signal through the model [34,35].

Each neuron in the network applies a weighted sum to its inputs, adds a bias, and then passes this sum through an activation function to produce an output. The mathematical representation of the output of a neuron can be given as:(11)$$f\left(\sum_{i=1}^{n}\omega_{i}x_{i}+b\right)$$
where $\omega_{i}$ are the weights, $x_{i}$ are the inputs, b is the bias, and f is the activation function.

Activation functions introduce non-linear properties to the network, enabling it to learn complex data patterns. Common activation functions include Sigmoid, Tanh, ReLU (rectified linear unit), and Softmax (for the output layer in classification problems) [36].

Backpropagation is used for training the network. It involves computing the gradient of the loss function with respect to each weight by the chain rule and propagating the error backward through the network from the output layer to the input layer. The update rule for a weight in the network is given by:(12)$$\omega_{new}=\omega_{old}-\eta \frac{𝜕L}{𝜕\omega }$$
where η is the learning rate and $\frac{𝜕L}{𝜕\omega }$ is the partial derivative of the loss L with respect to the weight ω.

The loss function measures the difference between the actual output and the predicted output of the network. Common loss functions include mean squared error (MSE) for regression problems and cross-entropy for classification problems. Optimization algorithms are used to minimize the loss function. Gradient descent is the most basic form, with variations like stochastic gradient descent (SGD), Adam, and RMSprop often used to improve convergence [37].

### 3.5. Hyperparameter Tuning

The “black box” problem in the context of hyperparameter tuning for neural networks refers to the challenge of understanding how changes in hyperparameters affect the model’s performance and outcomes. In machine learning models, especially in deep learning, there are numerous hyperparameters, such as the number of neurons in network layers, learning rate, regularization methods, batch size, activation functions, number of trees (n_estimators), and many others. Each of these hyperparameters can significantly impact the model’s effectiveness, its ability to generalize, and the computational time and complexity required for training [38].

Techniques and tools like grid search, random search, Bayesian optimization, evolutionary algorithms, and others can automatically explore the hyperparameter space to find optimal configurations. Tools that visualize changes (such as Matplotlib, Seaborn, Plotly, Weights & Biases, TensorBoard, Mlflow, Scikit-learn) [39], in the learning process and the impact of hyperparameters can help understand how specific settings affect the model [40].

In the ML and ANN models used, hyperparameter tuning was performed using the grid search method. This method is a hyperparameter optimization technique for machine learning models that involves systematically searching through a predefined space of hyperparameters. For each set of hyperparameters, the model is trained, and its performance is evaluated using a chosen methodology, often employing cross-validation. This process allows for the identification of the hyperparameter combination that yields the best model performance according to a specific criterion, such as accuracy. The main advantage of this method is its simplicity and ease of implementation, while its primary drawback is its high computational cost, especially with large hyperparameter spaces [41].

## 4. Results

### 4.1. Model Validation

An analysis of feature significance (the impact of independent variables) was conducted for three different models: random forest (RF), XGBoost (XGB), and multilayer perceptron neural network (MLP-ANN). For tree-based models, such as RF and XGB, feature importance is determined based on the impact of splits on these features within the tree structure, which is directly incorporated into these algorithms. This method assesses how much a split on a given feature reduces impurity in the decision tree nodes.

For neural networks, such as MLP-ANN, this analysis is more complex, as such models lack an inherent method for assessing feature importance. Instead, techniques like permutation feature importance (PFI) are used, which involves evaluating the change in model performance after perturbing the data for a given feature [42]. All results are summarized in Table 4. In the random forest model, the distribution of importance between tool rotational speed and welding speed is almost equal, with a marginal preference given to welding speed (50.79% compared to 49.21%). This suggests that both parameters are nearly equally influential in determining the outcome in the RF model. The XGBoost model, however, places a more significant emphasis on tool rotational speed (62.46%) over welding speed (37.54%), indicating that, for the XGBoost model, the speed of the tool’s rotation is a more critical predictor than the welding speed. For the MLP-ANN model, after normalizing the feature importance values, tool rotational speed accounts for 44.53%, while welding speed is slightly higher at 55.47%. The closer normalized values suggest that both parameters are of comparable importance to the MLP-ANN model’s predictions.

In the context of the study, where the analysis of feature significance was thoroughly conducted using random forest (RF), XGBoost (XGB), and multilayer perceptron neural network (MLP-ANN) models, the application of ANOVA (analysis of variance) is deemed not necessary. The comprehensive insights derived from these machine learning models, which demonstrated the near-equal importance of both independent variables—tool rotational speed and welding speed—on the dependent variable (UTS), underscore the adequacy of the methods employed. Given the complexity and the non-linear interactions between features captured by these models, ANOVA, which is traditionally used for comparing means across groups and assumes linear relationships, might not provide additional or more insightful information in this specific case [43].

The scatter plots shown in Figure 6 provide a visual comparison of the actual ultimate tensile strength (UTS) values with those predicted by three different machine learning models: random forest (RF), XGBoost (XGB), and multilayer perceptron (MLP).

The determination coefficient $R^{2}$, which measures the proportion of the variance for the actual UTS that is predictable from the input variables, ranges from 0.91 to 0.94 across the models. This high $R^{2}$, value suggests that all models—RF, XGB, and MLP—have a very good ability to explain the variability of the actual UTS values. The models are quite comparable in their predictive performance, with only a slight edge for the RF and XGB models over the MLP model, as indicated by their $R^{2}$ values.

The charts presented in Figure 7 illustrate the comparison of root mean squared error (RMSE) for three different machine learning models—MLP regressor, random forest, and XGBoost—as a function of the number of training samples. The graphs demonstrate how the RMSE changes with an increasing number of samples for both the training and test sets, which allows an assessment of how well each model generalizes to unseen data.

Analyzing the root mean squared error (RMSE) charts for three different machine learning models presents a comprehensive view of their performance across varying numbers of training samples. Figure 7a depicts the RMSE values for a random forest model. One observes that both training and test RMSE values decrease as the number of training samples increases. Notably, the test RMSE closely follows the training RMSE, suggesting that the model generalizes well. The RMSE values reach a level of stability after a certain number of training samples, indicating that the model achieves a state of equilibrium, where additional training samples do not significantly improve the model’s performance on the test set. Figure 7b presents the performance of an XGBoost model. Initially, the test RMSE is high but quickly converges towards the training RMSE with more training samples. A rapid decline in test RMSE with an increase in training samples suggests that the model benefits significantly from more data initially. The RMSE values for both training and test sets eventually stabilize, indicating a good fit without signs of overfitting or underfitting. Lastly, Figure 7c shows the RMSE values for an MLP-ANN model. Here, the test RMSE starts at a much higher value than the training RMSE, indicating potential overfitting when the number of training samples is limited. However, as more training samples are added, the test RMSE decreases, suggesting improved generalization. Despite this improvement, the test RMSE remains higher than the training RMSE throughout, which may imply that the model still overfits the training data to some extent.

In summary, the random forest model exhibits the best performance in terms of RMSE, with strong generalization from the outset and stability as the number of training samples increases. The XGBoost model also performs well, particularly after an initial period with fewer samples. The MLP-ANN model, while improving with more data, consistently shows higher error rates on the test set, indicating that it may not generalize as effectively as the other models.

The performance metrics in Table 5 and the *p*-value comparisons in Figure 8 collectively suggest that the random forest (RF), extreme gradient boosting (XGB), and multilayer perceptron (MLP-ANN) models have similar levels of predictive capabilities. The R^2^ values, which reflect the proportion of variance explained by the model, are high for all, with RF and XGB showing marginally higher values than MLP-ANN. For the mean absolute error (MAE) and mean absolute percentage error (MAPE), XGB registers the lowest values, indicating its predictions are, on average, closest to the actual values. When considering the mean squared error (MSE) and root mean squared error (RMSE), which square the prediction errors before averaging—thereby giving greater weight to larger errors—XGB again has the lowest values. This demonstrates that XGB is the most robust to large errors, directly contrasting with any previous suggestion that it might produce larger errors. These metrics show that XGB not only makes smaller errors on average but is also less likely to make large errors compared to the other models. Accuracy percentages are high across all models, with XGB having a slight advantage. However, the *p*-values depicted in Figure 8 indicate that the differences in model performance metrics are not statistically significant, as they all exceed the 0.05 threshold for significance. This implies that the observed marginal advantages of XGB in Table 5 metrics do not translate into statistically significant superiority over RF and MLP-ANN. Therefore, with no significant statistical difference in performance, selection among these models for practical application would likely be influenced by factors beyond the metrics presented, such as computational efficiency, ease of model tuning, and interpretability.

### 4.2. Optimization and Extrapolation of FSW Process Parameters

The created ML and ANN models can be used to optimize the parameters of the FSW process. The main criterion for optimization is to select a pair of technological parameters, namely tool rotation and welding speed, so that the UTS of the joined sheet is as close as possible to the UTS of the base material. This approach is becoming increasingly popular among researchers. An example is the work of Tauqir Nasir et al. [36], where the authors emphasize the importance and potential of applying ML and ANN techniques in the optimization of FSW and FSSW processes, demonstrating how advanced data analysis techniques can contribute to improving quality and efficiency in industrial production. It has been shown that ML methods can predict the responses of the FSW process with an error below 5%. As shown in the previous paragraph, optimization can be applied to improve the hyperparameters of the ML and ANN models and to search for optimal process parameters according to the given criterion. For this purpose, an algorithm for finding optimal parameters in the analyzed input data set or extrapolating parameters beyond the analyzed range can be applied to the trained ML and ANN models. Extrapolating results beyond the range of studied data requires caution, as predictions may not always accurately reflect the actual behavior of materials and processes. When using extrapolation, it should be taken into account that the obtained results should be experimentally confirmed.

Below is the original pseudocode (Table 6) that captures the logic of the Python function. It describes the process of generating an extended range (by 50% on both sides) for two parameters, iterating through all combinations to find the one that yields the predicted UTS closest to the target UTS, and then returning those optimal parameters.

In this pseudocode, lines 02–04 are responsible for inputting the model, target value, and ranges for two parameters (Param1 and Param2), along with their minimum and maximum values and the number of steps. The loops in lines 07–17 iterate through all possible combinations of Param1 and Param2, predict the outcome using the model, and then compare it with the target value to find the best match. Lines 18–19 display the best-found parameters and the smallest difference between the predicted and target values.

In Table 7, the optimized and extrapolated parameters obtained from the performed ML and ANN models are presented. The optimizations were conducted using a systematic parameter search method and leveraging trained ML and ANN models. The search algorithm utilizes predictions from ML and ANN models to iteratively enhance welding parameters towards maximizing the objective function, i.e., maximizing UTS. Response surfaces illustrating the influence of process parameters on UTS were constructed based on the developed ML and ANN models (Figure 9). Machine learning models RF and XGB present highly similar predictions. However, the MLP-ANN model exhibits certain differences despite employing regularization techniques. This may be due to decision trees performing significantly better with small input datasets (in our case, only 9 samples with 5 repetitions) compared to MLP-ANN. Additionally, the small dataset size contributes to a greater susceptibility to overfitting for MLP-ANN, as can be observed on the learning curve (Figure 7c), which is associated with their more complex structure. MLP may be more sensitive to the presence of nonlinearity in the data, whereas decision trees may better handle nonlinear dependencies due to their recursive nature of partitioning data into smaller subsets [44,45]. The differences in model performance for predicting the properties of FSW joints can be attributed to how each model handles the underlying complexity of the data, the balance between bias and variance, and their inherent algorithmic strengths and weaknesses. The optimal model choice depends on the specific requirements of the FSW process, the available data, and the desired balance between prediction accuracy and model interpretability.

### 4.3. Confirmation Test

Confirmation tests were conducted for the extrapolated parameters to validate the predictions of the ultimate tensile strength (UTS). Figure 10 displays a box plot comparing the strength tests to those of the base material, illustrating the outcomes of extrapolated FSW welding parameters in terms of joint strength. The parameters identified by the RF model (990 rpm and 175 mm/min) exhibited the highest strength among all tested, with a moderate standard deviation among the samples. Conversely, parameters determined using the XGBoost method showed slightly inferior performance, and the spread of results was considerable. The MLP-ANN method yielded the poorest outcomes, producing the weakest joint but with greater uniformity, as indicated by a very low standard deviation of results. Additionally, it’s noteworthy that the range of FSW process parameters extrapolated by the MLP-ANN model extended far beyond the analyzed range, making experimental verification crucial. The interpolation of FSW welding parameters using trained RF and XGB models did not significantly deviate from the original parameter matrix area. In the case of the MLP-ANN model, the deviation beyond the initial range was significant and failed to achieve the fully anticipated effects, despite predictions indicating a strength comparable to the base material.

The variations in parameters for extrapolation, arise from the distinct predictive outcomes generated by each model. This methodological approach is driven by the aim of exploring how far the parameters could be optimized to enhance the ultimate tensile strength (UTS) towards the target value of the base material’s strength. The differences in extrapolated parameters underscore the unique analytical capabilities of each model—random forest, XGBoost, and MLP-ANN—in navigating beyond the initial study range.

## 5. Conclusions

The presented studies have revealed the potential of advanced ML and ANN methods for significantly improving the FSW process through the optimization of technological parameters, even when faced with the challenge of a limited amount of input data. Based on the conducted research, the following conclusions have been drawn:The study applied ML and ANN methods, such as RF, XGBoost, and MLP, to optimize FSW parameters for 2024-T3 aluminum alloy, achieving very good results.RF improved variance reduction and overfitting control through bootstrap sampling, enhancing model diversity and performance.XGBoost enhanced performance with gradient boosting and regularization, handling missing data well in small datasets.MLP models achieved over 98% accuracy despite their complexity and potential overfitting issues, comparable to RF and XGBoost.An extrapolation algorithm developed in the study identified the optimal FSW parameters, with the RF model yielding the most accurate and consistent results.XGBoost and MLP-ANN were less precise in their predictions.The study successfully demonstrated the feasibility of using RF, XGBoost, and MLP-ANN to optimize and predict FSW joint mechanical properties with a small test set.RF yielded the best extrapolation results, emphasizing its effectiveness in optimizing production processes.Analysis of two independent parameters, rotation and feed, found both to be statistically significant to the outcome.Optimization results were consistent across all three models, with the resulting FSW joint strength at 93% relative to the base material.The statistical analysis of the differences in performance metrics between the models indicated that the discrepancies in their performance are too minor to be considered statistically significant.

## 6. Future Research Directions

Future research could be expanded to include the development of an image analysis-based system for real-time assessment of weld quality. This approach would leverage advanced imaging techniques to monitor the welding process and detect defects as they occur, enabling immediate corrective actions. Another direction could involve analyzing the force exerted during welding as a real-time indicator of weld quality. This method would measure the forces involved in the welding process to predict the quality of the weld, offering a novel way to ensure the integrity and strength of the joints as they are formed. Both approaches aim to enhance the control and consistency of the friction stir welding process, potentially revolutionizing the way weld quality is managed in real-time.

## Figures and Tables

**Figure 1 materials-17-01452-f001:**
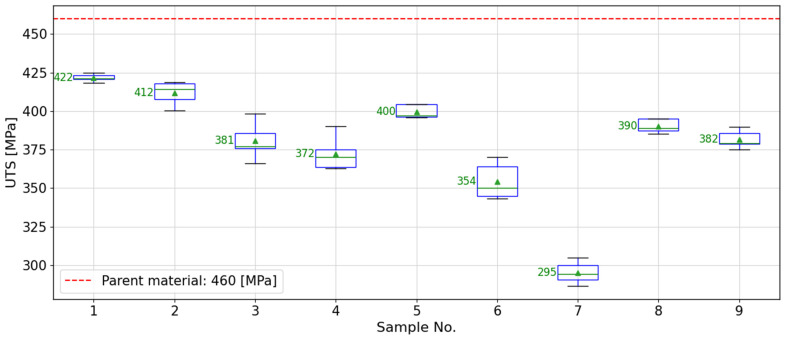
Box plot representation of the ultimate tensile strength (UTS) test results for friction stir welded butt joints of 1.6 mm thick 2024-T3 aluminum alloy sheets. The plot illustrates the spread of results across five specimens prepared from each welded sample, showcasing the variability in UTS measurements.

**Figure 2 materials-17-01452-f002:**
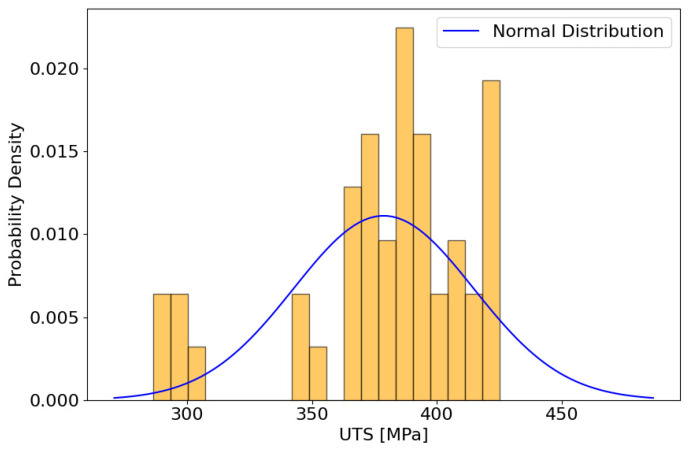
Histogram of ultimate tensile strength (UTS) values with a normal distribution curve.

**Figure 3 materials-17-01452-f003:**
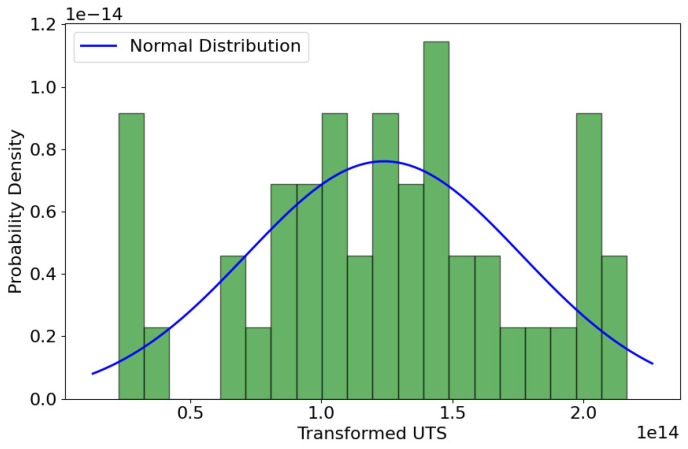
Box–Cox transformed ultimate tensile strength (UTS) values histogram with normal distribution curve.

**Figure 4 materials-17-01452-f004:**
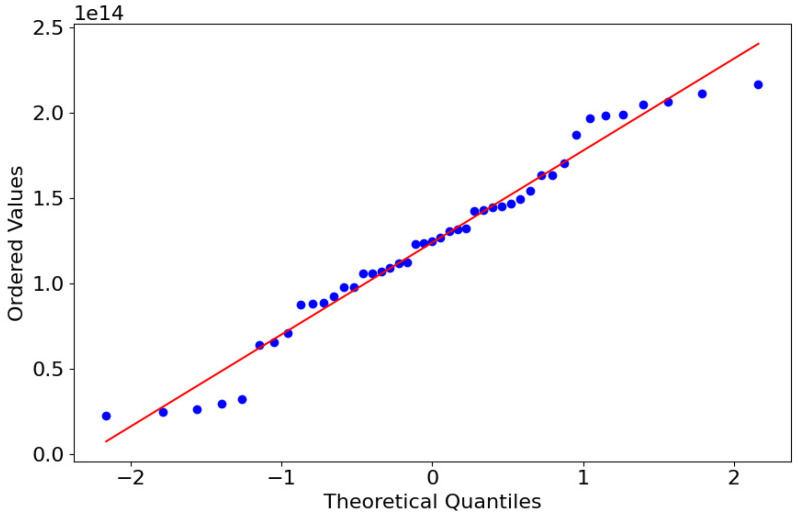
Q–Q plot of Box–Cox transformed ultimate tensile strength (UTS) values. The plot compares the ordered UTS values with the theoretical quantiles of a normal distribution line.

**Figure 5 materials-17-01452-f005:**
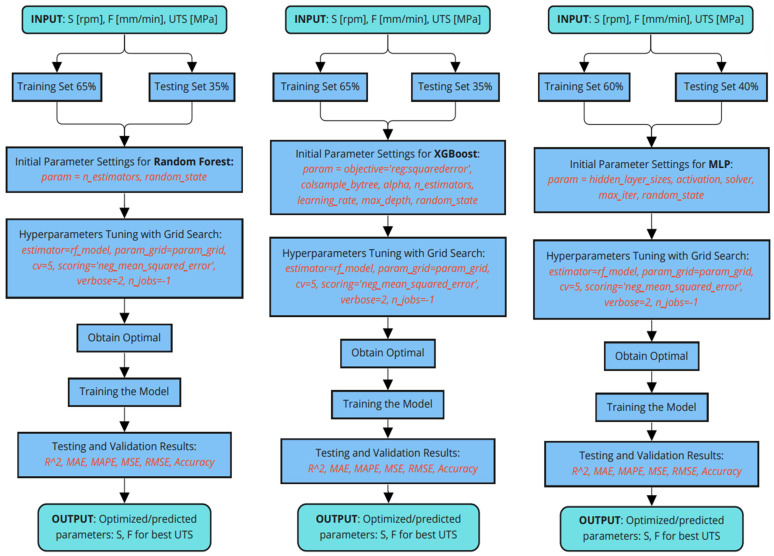
Flow chart of the ML and ANN prediction models.

**Figure 6 materials-17-01452-f006:**
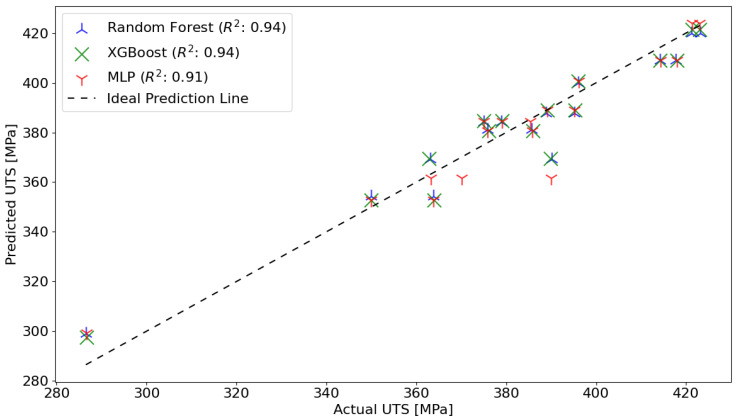
Scatter plot comparison of actual vs. predicted UTS using an RF, XGBoost, and MLP-ANN model.

**Figure 7 materials-17-01452-f007:**
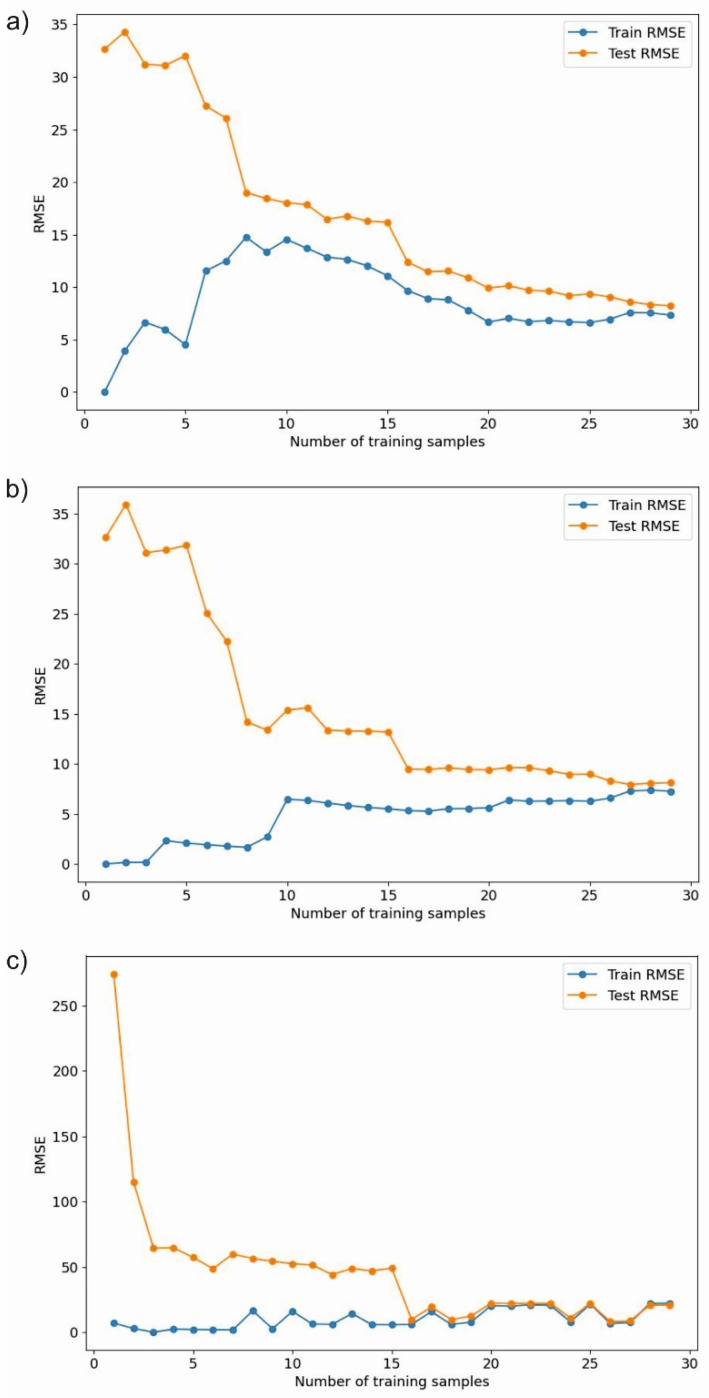
Comparative analysis of RMSE for machine learning models: (**a**) random forest, (**b**) XGBoost, and (**c**) MLP-ANN.

**Figure 8 materials-17-01452-f008:**
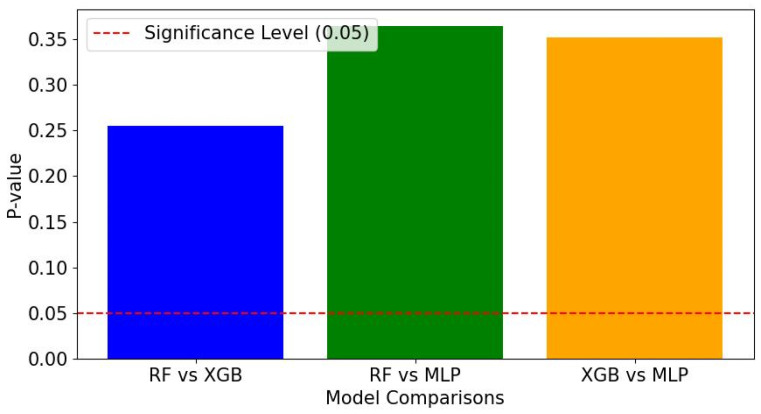
The comparison of *p*-values for predictive models indicates no statistically significant differences between random forest (RF), extreme gradient boosting (XGB), and multilayer perceptron (MLP) methods at the adopted significance level of 0.05.

**Figure 9 materials-17-01452-f009:**
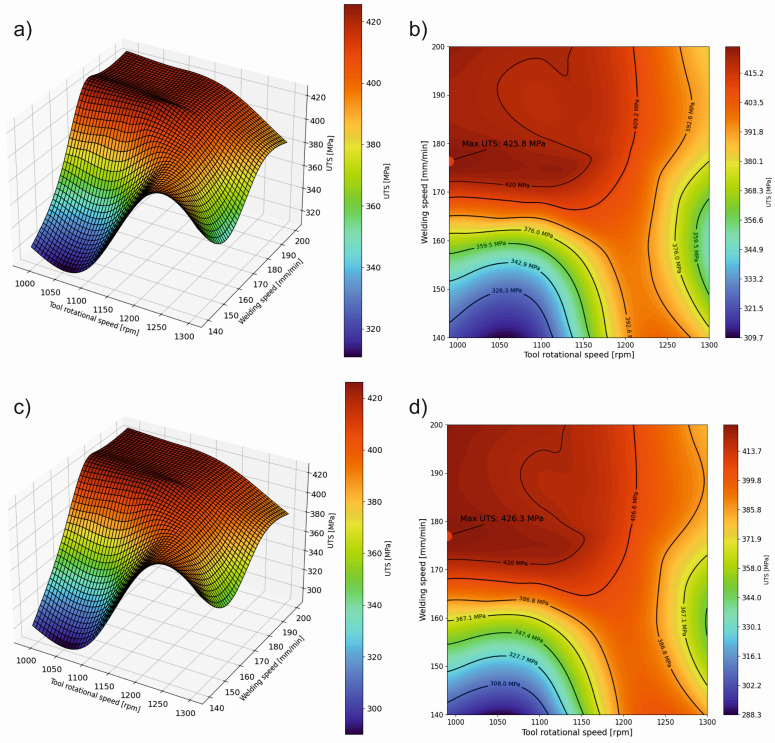
Response surface plots (**a**,**c**,**e**) and contour plots (**b**,**d**,**f**) generated by ML and ANN models for forecasting UTS in relation to FSW parameters (**a**,**b**) for the random forest model, (**c**,**d**) for the XGBoost model, (**e**,**f**) and for the multilayer perceptron MLP model.

**Figure 10 materials-17-01452-f010:**
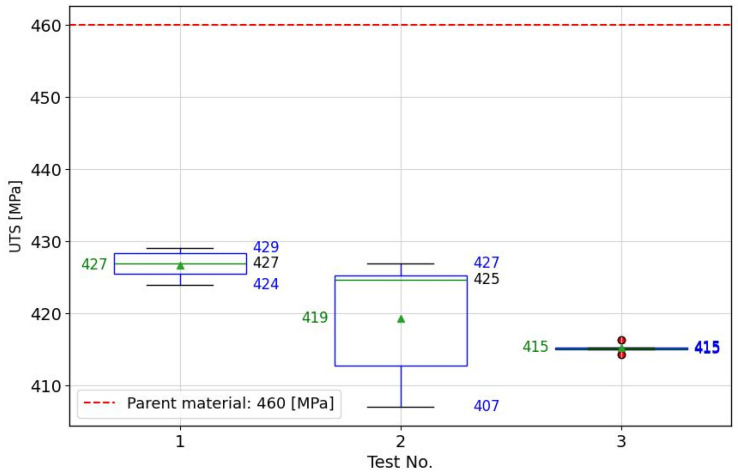
The UTS experimental validation results for interpolated FSW parameters by ML and ANN models. Parameters for test no. 1 (RF): tool rotational speed: 990 [rpm], welding speed: 175 [mm/min]. Parameters for test no. 2 (XGB): tool rotational speed: 1100 [rpm], welding speed: 170 [mm/min]. Parameters for test no. 3 (MLP): tool rotational speed: 700 [rpm], welding speed: 125 [mm/min].

**Table 1 materials-17-01452-t001:** Geometric parameters of the FSW tool and welding conditions.

Tool Parameters	Value	Tool View
Shoulder diameter D [mm]	12	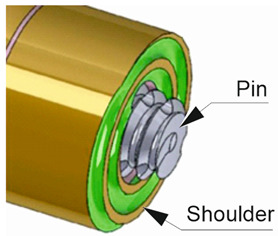
Pin diameter d [mm]	4.5	
Pin height [mm]	1.45	
Tool offset [mm]	0.05	
Dwell time [s]	10	
Tool tilt angle	0°	
Tool plunge speed [mm/min]	2	
Shoulder profile	Flat with spiral groove	
Pin profile	Cylindrical threaded	
D/d ratio of the tool	2.7	
Tool material	H13 steel	

**Table 2 materials-17-01452-t002:** Ultimate tensile strength UTS measurements and predictions.

Sample No.	Tool Spindle Speed [rpm]	Welding Speed [mm/min]	Measured UTS [MPa]	Mean Measured UTS [MPa]	Standard Deviation [MPa]	UTS Predicted RF [MPa]	UTS Predicted XGB [MPa]	UTS Predicted MLP [MPa]
1.	1100	180	418	422	2.44	421	421	421
			421					
			425					
			423					
			421					
2.	1200		408	412	7.68	410	409	413
			418					
			419					
			414					
			401					
3.	1300		398	381	12.06	380	381	381
			377					
			376					
			386					
			366					
4.	1100	160	370	364	11.07	368	370	369
			363					
			363					
			375					
			390					
5.	1200		405	400	4.39	400	401	401
			405					
			397					
			396					
			396					
6.	1300		350	364	11.84	353	353	353
			364					
			370					
			345					
			343					
7.	1100	140	300	304	7.35	298	298	298
			291					
			287					
			294					
			305					
8.	1200		395	390	4.57	386	389	389
			385					
			395					
			387					
			389					
9.	1300		390	382	5.85	385	388	388
			375					
			386					
			379					
			379					

**Table 3 materials-17-01452-t003:** Analysis of variance (ANOVA) for independent variables (tool rotation speed and welding speed) and the dependent variable (ultimate tensile strength—UTS).

Source	Sum Square	df	Mean Square	F-Value	*p*-Value
Model	49,978.34	5	9995.67	48.49	<0.0001 (significant)
A: Tool rpm	629.23	1	629.23	3.05	0.0885
B: Welding Speed	18,022.82	1	18,022.82	87.44	<0.0001 (significant)
AB	20,267.21	1	20,267.21	98.33	<0.0001 (significant)
A^2^	10,834.81	1	10,834.81	52.56	<0.0001 (significant)
B^2^	224.27	1	224.27	1.09	0.3033
Residual	8038.84	39	206.12		
Lack of Fit	5632.83	3	1877.61	28.09	<0.0001 (significant)
Pure Error	2406.01	36	66.83		
Cor Total	58,017.18	44			

**Table 4 materials-17-01452-t004:** Feature importance analysis for tool rotational speed and welding speed across machine learning models.

Model	Tool Rotational Speed [%]	Welding Speed [%]
Random Forest	49.21	50.79
XGBoost	62.46	37.54
MLP-ANN	44.53	55.47

**Table 5 materials-17-01452-t005:** Performance metrics for machine learning models.

Model	$R^{2}$	MAE	MAPE [%]	MSE	RMSE	Accuracy [%]
RF	0.94	6.80	1.83	67.61	8.22	98.17
XGBoost	0.94	6.47	1.74	65.87	8.12	98.26
MLP-ANN	0.91	6.59	1.77	83.69	8.42	98.23

**Table 6 materials-17-01452-t006:** Optimization of FSW parameters through extrapolation.

Pseudocode:
01. START 02. INPUT: Model (RF or XGB or MLP), TargetValue 03. INPUT: Range for Param1 (min, max, steps) 04. INPUT: Range for Param2 (min, max, steps) 05. Initialize BestDiff as INFINITY 06. Initialize BestParams as (0, 0) 07. FOR each value of Param1 in Range (Param1_min * 0.5, Param1_max * 1.5, steps): 08. FOR each value of Param2 in Range (Param2_min * 0.5, Param2_max * 1.5, steps): 09. TestParams = [Param1, Param2] 10. Prediction = MODEL.PREDICT (TestParams) 11. Diff = ABS (Prediction − TargetValue) 12. IF Diff < BestDiff THEN: 13. BestDiff = Diff 14. BestParams = TestParams 15. END IF 16. END FOR 17. END FOR 18. OUTPUT: ‘Best Parameters:’, BestParams 19. OUTPUT: ‘Smallest Difference:’, BestDiff 20. END

**Table 7 materials-17-01452-t007:** Comparison of optimized and extrapolated parameters for friction stir welding (FSW) as predicted by the random forest (RF), extreme gradient boosting (XGBoost), and multilayer perceptron (MLP) models.

Test No.	Model	Optimization Parameters [rpm]@[mm/min]	Mean Measured UTS [MPa]	Extrapolation Parameters [rpm]@[mm/min]	Mean Measured UTS [MPa]
1.	RF	1100@175	422	990@175	427
2.	XGBoost	1100@175		1100@170	419
3.	MLP-ANN	1100@180		700@125	415

## Data Availability

The data presented in this study are available on request.

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
