# Peer review of "Optimization of 2024-T3 Aluminum Alloy Friction Stir Welding Using Random Forest, XGBoost, and MLP Machine Learning Techniques"

_materials, 2024, doi:10.3390/ma17071452_

Round 1
Reviewer 1 Report
Comments and Suggestions for Authors
This paper studies how to enhance the friction stir welding of 2024-T3 aluminum alloy using machine learning. In particular, the authors tested the performance of artificial neural network and found interesting efficiency. This is an interesting study, with the topics suitable for the journal. The results are of good significance and novelty. Before it is acceptable, I only have several minor concerns that should be carefully addressed by the authors.
Therefore, I recommend that this nice piece of work is acceptable for the journal after a minor revision.
1. The title should be revised. Machine learning already contains neural network approaches.
2. The data used for modeling should be further discussed. In particular, some statistical analysis should be performance to show the data diversity.
3. For comparing the performance of different machine learning methods, I strongly suggest the authors to discuss how to properly compare different machine learning models, such as: https://doi.org/10.3390/pr7030151
4. There are several minor typos in the manuscript. Please double-check the paper.
Reviewer 2 Report
Comments and Suggestions for Authors
After carefully reviewing this study, I have come to understand that the structure of this manuscript is not suitable and should be rearranged by the authors if they intend to publish it. There are numerous minor errors in writing and scientific content. However, I will highlight some of my comments and issues below. If the authors can address them, the manuscript may be considered for publication:
-
The keywords should be separated by semicolons.
-
The introduction is incomplete and poorly structured by the authors. For instance, using "FSW" at the beginning of sentences is not ideal; it's better to fully write it out the first time it's mentioned (excluding the abstract). Additionally, the authors begin by explaining the history of FSW, but the manuscript focuses on modeling. Therefore, more information on machine learning and modeling in this material should be included.
-
The structure of the manuscript could be improved for better reader understanding. For example, compare it with the structure of other manuscripts such as those found at the following links: https://www.mdpi.com/2227-7390/11/13/3022 https://link.springer.com/article/10.1007/s00170-023-11728-z
-
The last paragraph of the introduction is incomplete, and the novelty of the word and the motivation are unclear. Perhaps it would be better to move Section 2 from the first paragraph to the last paragraph of the introduction.
-
References and standards for the selection of the material should be added to Table 1.
-
Each part should be mentioned or indicated in the tool view.
-
Table 2 needs standard deviations, errors, and the unit should be in MPa. Clarify whether "RPM" or "rpm" should be used.
-
In Figure 1, check units ([rpm]/[mm/min]); it's recommended to use only one unit. Also, increase the quality of the figures.
-
Are there real UTS strength values for comparison? If so, provide information about the tests, machine, and standards in Section 2.
-
Tables should be named on pages 7 and 8. Equations, tables, and figures should be cited in-text and explained.
-
Determine if it's necessary to include the pseudocode.
-
Avoid using "+/-"; use "(±)" instead.
-
In Figure 3, check the name and explanation. Are there normal plots for this data? Indicate the red, green points, and black dashed line.
-
Clarify if the significance level is greater than 0.05 or less than it in Figure 5. Are all points out of significance?
-
For Figure 6, label different plots separately (surface plot from contour plots). Ensure the plots align with the first input parameters mentioned. It seems there are data points beyond the specified welding speeds, which may invalidate the RSM without experimental data.
-
Rewrite the conclusion as bullet points for easier reader understanding.
-
Add more references to emphasize the importance of the topic and make it easier for other researchers to find.
Comments on the Quality of English Language
some small wordy errors.
Reviewer 3 Report
Comments and Suggestions for Authors
The paper can be accepted, but it is necessary to carry out certain refinements in order to improve the quality of the paper and its application for subsequent tests.
It is necessary to review and cite an even greater number of recent papers from this and similar fields, such as:
N. Ratković, Ž. Jovanović Pešić, D. Arsić, M. Pešić, D. Džunić, Tool geometry effect on material flow and mixture in FSW, Advanced Technologies & Materials, Vol.47, No.2, pp. 33-36, ISSN 2620-0325, Doi 10.24867/ATM-2022-2-006, 2022.
https://doi.org/10.2478/msp-2022-0013
Expand the introduction or other chapters with papers from this field.
At the end of the introduction, state what is the main contribution of the paper and how does this paper differ from similar papers in this field? What is the reason, i.e. why should this paper be published?
Could you provide a clearer explanation regarding the rationale behind selecting the specific parameter levels as depicted in table 2, specifically the tool spindle speeds of 1100, 1200, and 1300 rpm, and welding speeds of 140, 160, and 180 mm/min?
In Table 2 and in the text, standardize MPa instead of Mpa.
How did the used models interpret the importance of individual variables, especially tool rotation speed and welding speed, in relation to the final mechanical characteristics of the joints?
Explain in more detail how the Random Forest, XGBoost and MLP-ANN models showed different performances in predicting the mechanical properties of Friction Stir Welding joints on aluminum alloy 2024-T3, what factors most influenced the differences in the performance of these models?
What are the implications of these results for the practical application of the Friction Stir Welding (FSW) process in industry, especially with regard to the optimization of the process parameters?
Why did the authors in Table 5 not use the same parameters for all models for extrapolation? Was it to facilitate a more meaningful comparison among the models, or was it deemed unimportant?
How could further research be expanded based on the results and conclusions of this research, and how could the predictive power of the model be improved in the future?
Increase the number of references in the paper based on the given remarks.
Analysis of the research results, which was carried out in detail using Random Forest, XGBoost and Multilayer Perceptron Neural Network models, provides deep insights into the significance of features. The authors have successfully applied various techniques for assessing the significance of features in accordance with the inherent characteristics of each model.
This research recognizes the importance of model selection for practical application, which could be influenced by factors such as computational efficiency, ease of model fitting, and interpretability, given the lack of statistically significant differences in model performance.
Round 2
Reviewer 2 Report
Comments and Suggestions for Authors
There is no more comments.